# Preimplantation Genetic Testing for Aneuploidy (PGT-A) Reveals High Levels of Chromosomal Errors in In Vivo-Derived Pig Embryos, with an Increased Incidence When Produced In Vitro

**DOI:** 10.3390/cells12050790

**Published:** 2023-03-02

**Authors:** Reina Jochems, Carla Canedo-Ribeiro, Giuseppe Silvestri, Martijn F. L. Derks, Hanne Hamland, Louisa J. Zak, Egbert F. Knol, Alan H. Handyside, Eli Grindflek, Darren K. Griffin

**Affiliations:** 1Norsvin SA, 2317 Hamar, Norway; 2School of Biosciences, University of Kent, Canterbury CT2 7NH, UK; 3Topigs Norsvin Research Center, 6641 SZ Beuningen, The Netherlands; 4Animal Breeding and Genomics, Wageningen University & Research, 6700 AH Wageningen, The Netherlands

**Keywords:** cytogenetics, porcine, in vitro embryo production, aneuploidy, parthenogenetic, androgenetic

## Abstract

Preimplantation genetic testing for aneuploidy (PGT-A) is widespread, but controversial, in humans and improves pregnancy and live birth rates in cattle. In pigs, it presents a possible solution to improve in vitro embryo production (IVP), however, the incidence and origin of chromosomal errors remains under-explored. To address this, we used single nucleotide polymorphism (SNP)-based PGT-A algorithms in 101 in vivo-derived (IVD) and 64 IVP porcine embryos. More errors were observed in IVP vs. IVD blastocysts (79.7% vs. 13.6% *p* < 0.001). In IVD embryos, fewer errors were found at blastocyst stage compared to cleavage (4-cell) stage (13.6% vs. 40%, *p* = 0.056). One androgenetic and two parthenogenetic embryos were also identified. Triploidy was the most common error in IVD embryos (15.8%), but only observed at cleavage, not blastocyst stage, followed by whole chromosome aneuploidy (9.9%). In IVP blastocysts, 32.8% were parthenogenetic, 25.0% (hypo-)triploid, 12.5% aneuploid, and 9.4% haploid. Parthenogenetic blastocysts arose from just three out of ten sows, suggesting a possible donor effect. The high incidence of chromosomal abnormalities in general, but in IVP embryos in particular, suggests an explanation for the low success of porcine IVP. The approaches described provide a means of monitoring technical improvements and suggest future application of PGT-A might improve embryo transfer success.

## 1. Introduction

Chromosomal errors in embryos are a major cause of implantation failure, spontaneous abortions, and congenital birth defects in humans. Preimplantation genetic testing for aneuploidy (PGT-A) is frequently (but sometimes controversially) used in IVF embryos as a fertility treatment to identify chromosomal abnormalities and select euploid embryos prior to transfer [1,2]. In cattle, embryo production (either in vitro-produced (IVP) or in vivo-derived (IVD)) is employed for breeding purposes, e.g., to shorten generation intervals and facilitate the dissemination of genetics from elite livestock. These embryos are often analyzed for genomic estimated breeding values but, to date, rarely for their chromosomal constitution [3,4]. Characterization of chromosomal errors is of interest as they directly affect fertility by reducing the pregnancy rate or number of viable offspring [5]. They can arise in the embryo before fertilization in the gametes (meiosis), during fertilization (e.g., polyspermy), or after fertilization (mitosis), which can lead to mosaicism (presence of euploid and abnormal cells in the same individual) [6]. If applied correctly, PGT-A algorithms can be useful to detect a range of chromosomal abnormalities, such as monosomy (missing whole chromosome), trisomy (extra whole chromosome), segmental error (where only part of the chromosome is affected), uniparental disomy (UPD, when both chromosomes of a pair are inherited from only one parent), polyploidy of and variations thereof (hypo- and hyperpolyploidy), haploidy (a single set of chromosomes), parthenogenesis (diploid with only maternal genome), and androgenesis (diploid with only paternal genome) [7,8]. Recently, it has been demonstrated in cattle that selection against aneuploid embryos using PGT-A algorithms can improve overall pregnancy and live birth rates by 7.5% and 5.8%, respectively [9]. In pigs, it has been suggested that aneuploidy is not a prevalent cause of embryo mortality in vivo [10], but any loss is undesirable as it could result in a reduced litter size. Additionally, the efficiency of porcine IVP is often limited by an inconsistent blastocyst yield in each IVP round. Characterization of chromosome abnormality and its origin in porcine embryos could therefore provide a better understanding of its relevance in our IVP system and in vivo embryo loss.

Prior studies on cattle, sheep, horse, and pig suggested that there is a higher incidence of aneuploidy (extra or missing whole chromosomes) in IVP compared to IVD embryos [11,12,13,14]. The most used older technologies to assess aneuploidy are fluorescent in situ hybridization (FISH) and comparative genomic hybridization (CGH). The first allows to visualize chromosomes through hybridization with specific fluorescent labelled DNA sequences called probes, while the later uses the genomic DNA of the individual of interest compared to an euploid reference to detect alterations in copy number. However, CGH is unable to detect triploidy robustly and FISH is limited in the number of chromosomes it can analyze. Neither are able to detect UPD, parthenogenesis, nor androgenesis. Use of single nucleotide polymorphism (SNP)-based PGT-A algorithms can characterize the incidence and origin of all chromosomal errors in mammalian (including IVD and IVP porcine) embryos. This is an attractive procedure as the use of SNP microarrays is affordable and often the modality used by breeding companies to evaluate the genetic merit of individual embryos by calculating genomic estimated breeding values. To our best knowledge, the panoply of chromosomal errors has not been assessed systematically in porcine embryos and thus the issue of whether there are differences between early and late-stage embryos and IVD vs. IVP embryos remains uncertain. To this end, in this study we tested the hypothesis that IVD embryos have a lower incidence than IVP and that blastocysts have fewer chromosome errors than cleavage stage embryos (as in humans [15]).

## 2. Materials and Methods

### 2.1. Study Design

In vivo-derived embryos (*n* = 123) were collected, and in vitro blastocysts (*n* = 68) were produced between March 2020 and August 2021. Oocytes and embryos were collected from sows originating from Nucleus farms that were routinely slaughtered because of their breeding value, not because of sub/infertility. A tissue sample from each sow was stored at −80 °C for genotyping for identification. The DNA extraction from the ovaries was performed by BioBank, Hamar and genotyping was performed at CIGENE, Norwegian University of Life Sciences, Ås, using the Illumina GeneSeek custom 25K SNP chip (Lincoln, NE, USA).

### 2.2. In Vivo Embryo Collection

Two nulli- and three primiparous Landrace sows were inseminated with fresh semen from the same Landrace boar. The animals were culled at day 4, 5, or 6 of the cycle (day 0 is the onset of oestrus) and embryos were collected from the uterine tracts by flushing the top part of the horn. Briefly, 30 mL of prewarmed TL-HEPES-PVA at 37 °C was inserted with a syringe and needle at the side of the oviduct and approximately 40 cm of the uterine horn was flushed, after which the medium was collected in a 50 mL tube. This was repeated twice to collect most embryos. The number of embryos and embryo stages were recorded and the number of corpora lutea were counted on the ovaries to calculate the collection rate (Table 1).

### 2.3. In Vitro Embryo Production

In vitro embryo production was carried out as previously described [16]. Porcine X medium (PXM) was used for washing cumulus-oocyte complexes, porcine oocyte medium (POM) for maturation, porcine gamete medium (PGM) for fertilization, and porcine zygote medium-5 (PZM-5) for embryo culture [17]. Briefly, follicular phase ovaries were collected 20–24 h after weaning from 10 Landrace sows (parity 1 and 2) and transported to the laboratory in 0.9% NaCl at 25–30 °C. All 3–8 mm follicles were aspirated four hours after slaughter using an 18-gauge needle and 10 mL syringe, after which cumulus-oocyte complexes (COCs) with a compact cumulus and evenly granulated cytoplasm were washed in PXM and matured in modified POM. For the first 20 h, COCs were matured in POM supplemented with 0.05 IU/mL porcine FSH and LH (Insight Biotechnology Ltd., Wembley, Middlesex, UK), and 0.1 mM dibutyryl-cAMP (dbcAMP) in a humified atmosphere containing 6% CO_2_ in air. Subsequently, COCs were matured for another 24 h in POM without hormones and dbcAMP. Oocytes from the different sows were not pooled, but matured, fertilized, and cultured per sow in groups of 25–35 per well of a Nunc^®^ four-well multidish containing 500 mL pre-equilibrated medium. Fertilization was performed with cryopreserved sperm from one Landrace boar and straws originated from the same ejaculate. Each 2.5 mL straw was thawed at 50 °C for 50 s and diluted in 40 mL Tri-X-cell (IMV technologies, L’Aigle, France) at room temperature (RT). Sperm cells were washed and selected at RT using Percoll^®^ density gradient centrifugation by layering 2 mL of 45% Percoll on top of 2 mL 90% Percoll. Finally, 1 mL of semen was carefully placed on top followed by centrifugation at 700× *g* for 20 min. Supernatant was removed by aspiration, the pellet was resuspended in 4 mL PGM without BSA and centrifuged at 500× *g* for 5 min. The pellet was then resuspended in 150–200 µL PGM without BSA. Sperm motility and concentration were measured by computer assisted sperm analysis (CASA) using a Sperm Class Analyzer^®^ version 6.1 (Microptic SL, Barcelona, Spain), equipped with a phase contrast Eclipse Ci-S/Ci-L microscope (Nikon, Japan) and Basler digital camera (Basler Vision Technologies, Ahrensburg, Germany). Spermatozoa were diluted to 5 × 10^5^ progressively motile sperm cells/mL in 300 µL pre-equilibrated PGM with BSA. Groups of 25–35 COCs were co-incubated at 3.0 × 10^4^ progressively motile sperm cells/mL. After 2 h of co-incubation, oocytes were transferred to a new well with 500 µL PGM medium to remove an excess of sperm cells. After a total of 4 h fertilization, presumptive zygotes were denuded of cumulus cells by vortexing for 1 min in 2 mL PXM in a 15 mL tube. The presumptive zygotes were washed twice in PXM medium and once in PZM-5 before culture in 500 µL PZM-5 under 400 µL mineral oil (IVF Biosciences, Falmouth, UK) at 38.9 °C in an humified atmosphere containing 6% CO_2_ and 7% O_2_. At day 4 of culture (fertilization day 0), PZM-5 medium was replaced with 250 µL new equilibrated PZM media. Embryos were cultured until day 5 or 6 when they reached blastocyst stage.

### 2.4. Embryo Storage, Whole Genome Amplification and Genotyping

All embryos were incubated with 0.5% pronase in phosphate buffered saline (PBS) to remove the zona pellucida and DNA of any attached cumulus and/or sperm cells. In vivo-derived embryos were incubated for approximately 45–60 s, and in vitro-produced blastocysts for 30–45 s. Embryos were individually washed twice in PBS and once in REPLI-g Advanced Single Cell Storage buffer before they were individually stored in 3 µL storage buffer at −80 °C until further analysis. Samples were transported on dry ice to CIGENE, Norwegian University of Life Sciences, Ås, where the embryos were subjected to DNA extraction and whole genome amplification (WGA) using the REPLI-g Advanced DNA Single Cell Kit (Qiagen, Oslo, Norway) following the manufacturer’s instructions. Briefly, samples were denatured in 3 µL Denaturation Buffer by incubation at RT for 10 min, after which 3 µL Stop Solution was added. Samples were then kept on ice and 40 µL REPLI-g sc Master mix was added to 9 µL of denatured DNA after which the samples were amplified at 30 °C for 2 h, followed by inactivation of REPLI-g sc DNA Polymerase by incubation at 65 °C for 3 min. Amplified DNA was genotyped the same day using the Illumina GeneSeek custom 25K SNP chip. Only samples with a call rate ≥ 80% were included in the further study. After filtering, 101 IVD embryos and 64 IVP blastocysts were analyzed.

### 2.5. PGT-A Analysis

Based on the SNP genotypes, the chromosome analysis was conducted at the University of Kent, UK. Computing and PGT-A diagnosis were performed using a combination of three PGT-A algorithms per embryo [9]: signal intensity data B-Allele Frequency (BAF) and Log R Ratio (LRR) graphs, Karyomapping and Gabriel–Griffin plots. Briefly, BAF and LRR graphs were plotted using Circos software [18] to detect copy number variations across the karyotype [19], Karyomapping was employed to trace the parental origin (maternal or paternal) of each abnormality and to detect triploidy [20], and Gabriel–Griffin plots were employed to clarify the meiotic origin of trisomies (MI, MII, or mitotic) [21]. Karyomapping was used to classify abnormalities that result in a normal karyotype, as is the case of the uniparental disomy (UPD). Monosomies detected only in LRR/BAF plots but not by Karyomapping were classified as having originated de novo in the embryo (and therefore as being of mitotic origin). Examples of BAF and LRR graphs Karyomaps and Gabriel–Griffin plots are shown in Figure 1 and Figure 2 for some abnormal cases.

### 2.6. Statistical Analysis

The results are presented as percentages with their 95% confidence intervals (CI) for proportions unless stated otherwise. Proportions between different groups were analyzed by Fisher’s exact test and *p* ≤ 0.05 was considered statistically significant. Figures were plotted using GraphPad Prism version 9.0 (GraphPad Software, San Diego, CA, USA).

### 2.7. Data Availability

The data presented in this study are not publicly available due to being part of the intellectual property of Norsvin SA and Topigs Norsvin.

## 3. Results

### 3.1. Chromosomal Errors and Aneuploidy Classes

We found an overall incidence of chromosomal errors of 31.7% (*n* = 32/101) in IVD embryos. Fewer chromosomal errors were indicated at the blastocyst stage compared to the 4-cell stage (*p* = 0.056, Figure 3a). The levels were 40.0% in 4 cells (*n* = 10/25), 35.0% in 6–12 cells (*n* = 7/20), 35.3% in morulae (*n* = 12/34) and 13.6% in blastocysts (*n* = 3/22), respectively. The different chromosome abnormality classes found among the embryos in all development stages are shown in Figure 3b. A high incidence of triploid embryos (15.8%, *n* = 16/101) and embryos with aneuploidy (individual whole chromosome error) (9.9%, *n* = 10/101) was observed. This last class includes embryos showing monosomy, trisomy, and/or UPD. Triploid embryos were only observed at the early embryo developmental stages and not at the blastocyst stage (Figure 3c). Furthermore, two parthenogenetic embryos (only maternal genome) at the 12-cell and morula stage and one androgenetic embryo (only paternal genome) at the morula stage were identified.

In IVP blastocysts, only 20.3% were euploid (*n* = 13/64). A high incidence of parthenogenetic (32.8%, *n* = 21/64) and (hypo)triploid blastocysts (25.0%, *n* = 16/64) was indicated followed by 12.5% of the blastocysts having aneuploidy (individual whole chromosome error in the graph below) (*n* = 8/64) and 9.4% being haploid (*n* = 6/64) (Figure 4). Parthenogenetic blastocysts were only obtained from sow 6, 9, and 10, which originated from two different IVP rounds (Table 2). From sow 9, all 13 blastocysts were parthenogenetic, while only 44.4% (*n* = 4/9) and 57.1% (*n* = 4/7) of the embryos were parthenogenetic from sow 6 and 10, respectively. Furthermore, euploid blastocysts were obtained from 6 out of 10 sows, while only blastocysts with errors were produced from the other 4 sows.

The results indicate that IVP blastocysts have a higher incidence of errors (79.7%, *n* = 51/64) as compared to IVD blastocysts (13.6%, *n* = 3/22) (*p* < 0.001).

### 3.2. Origin of Chromosomal Errors

In total, 36 errors were observed in IVD embryos and 57 in IVP embryos (Table 3). Errors with a maternal origin were prevalent in IVP embryos (71.9%, *n* = 41/57), while errors from paternal origin where prevalent in IVD embryos (52.8%, *n* = 19/36) (*p* < 0.01). Furthermore, 2.0% (*n* = 2/101) and 4.7% (*n* = 3/64) of the errors originated from the embryo in IVD and IVP embryos, respectively. Tri- and hypotriploidy in both IVD and IVP embryos originated with a higher frequency from the paternal side (75.0%, *n* = 12/16), which is most likely related to polyspermy, compared to the maternal side (25.0%, *n* = 4/16) which is related to meiotic non-disjunction in the oocyte. All monosomies in both IVD and IVP embryos were of maternal origin. One case of uniparental disomy was identified in an IVP blastocyst on chromosome 18, which had a paternal origin. For trisomies, it was assessed whether the errors originated from meiosis I, meiosis II, or during embryonic development (mitotic error) (Figure 5). Most trisomies originated from meiosis I (50.0% in both IVD (*n* = 6/12) and IVP (*n* = 4/8) embryos). In IVD embryos, most trisomies originated from errors during paternal meiosis I (80%, *n* = 4/5), while no paternal trisomies were observed in the IVP embryos. The androgenetic in vivo embryo arose from a dispermic event, in which two sperm cells fertilized an empty oocyte.

### 3.3. Incidence of Whole Chromosomal Errors by Chromosome

The incidence of errors by chromosome are shown in Figure 6. A total of 28 whole chromosomal errors were found in IVD (*n* = 14) and IVP (*n* = 14) embryos. Chromosome 18 showed a higher incidence of errors (*n* = 3/14) in IVD embryos, as well as a higher frequency of trisomies (*n* = 3/12). Only chromosomes 2 and 14 presented monosomies, accounting for one each of the total of two monosomies.

Meanwhile, in IVP blastocysts, chromosome 14 was the most affected (*n* = 4/14), showing a similar incidence for monosomy and trisomy. This was followed by chromosomes 13 and 15 (both *n* = 3/14). Chromosome 13 presented more trisomies, whereas chromosome 15 presented more monosomies. Interestingly, the only UPD found in this dataset belonged to an IVP blastocyst on chromosome 18.

### 3.4. Chromosomal Errors by Sex

Excluding parthenogenetic, androgenetic, and haploid embryos, due to their genetic material being only from one parent, there was a higher prevalence of female embryos in IVD embryos (57.7%, *n* = 56/97, *p* = 0.04) but not in IVP embryos (51.4%, *n* = 19/37). Female and male embryos were affected by chromosomal abnormalities at similar rates, with 30.4% vs. 26.8% in IVD embryos and 78.9% vs. 50.0% (*p* = 0.09) in IVP blastocysts (Table 4).

A higher percentage of triploid embryos was female in both IVD (12.4%, *n* = 12/97) and IVP embryos (29.7%, *n* = 11/37) (Figure 7). Furthermore, tetraploid IVD embryos were only male. Whole chromosome errors were equally distributed across the sexes in both IVD and IVP embryos.

## 4. Discussion

The results presented suggest that there is a high prevalence of chromosomal errors in porcine (especially IVP) embryos, which potentially can be one of the reasons behind the inconsistency of IVP in establishing blastocyst rates. Although it might be argued that a high prevalence in multiparous animals is expected, the disparity between IVP and IVD embryos suggests that technical improvements in the IVP process need to be made before it mimics the in vivo situation and the benefits of porcine IVP can be fully realized. We establish proof of principle for the use of SNP-based data, applying algorithms that can monitor the progress of the improvement of the IVP process, and ultimately apply PGT-A to select chromosomally abnormal embryos as has been established in cattle [9]. However, this is practically still challenging in porcine as successful biopsy followed by vitrification and embryo survival has not yet been reported in pig embryos to our best knowledge. Currently, 25–30 fresh [22] or 40 vitrified [23] in vivo morulae or blastocysts are required for a successful gestation when performing non-surgical embryo transfer in pigs. The number of embryos required per transfer might be reduced and efficiency increased when it is possible to select euploid pig embryos prior to transfer.

We found an error incidence of 13.6% in IVD and 79.7% in IVP blastocysts, suggesting that the application of in vitro technologies might induce these errors. Previous separate studies in pig embryos show an incidence of 14.3–36.7% in in vivo [10,24] and 37–39.1% in vitro embryos [12,25]. Although in IVD embryos the incidence is similar, in our results, we found a higher incidence, which may be due to technique-induced diagnostic errors. The SNP-based algorithms allow us to detect balanced and unbalanced chromosomal changes, making them more powerful for diagnostics. The tendency of IVD embryos to present with less errors has also been suggested in other mammalian species such as cattle, horse, and sheep [11,13,14,26].

### 4.1. In Vivo-Derived Embryos

Our results indicated a lower incidence of chromosomal errors in IVD blastocysts (13.6%) compared to 4-cell stage embryos (40%). This is in line with the suggestion that chromosome abnormality is mostly responsible for early embryo mortality [5,10,27]. Griffin et al. recently established that aneuploid cells preferentially locate away from the developing fetus in human embryos and this is part of the self-correction mechanism as the embryo proceeds to blastulation [28]. Indeed, it is thought that the human embryo is chromosomally fluid at cleavage stage [15]. Similar studies in porcine embryos (and other mammalian embryos, such as cattle) would be of interest to understand whether this is transversal to other species or unique to humans.

Surprisingly, two parthenogenetic (12-cell and morula, diploid) and one androgenetic embryo (morula, diploid) were classified in the IVD embryos. To the best of our knowledge, this is the first time that parthenogenetic and androgenetic embryos have been characterized in in vivo-derived porcine embryos. The overall error incidence in the present study was higher compared to results from others who reported 14.3% aneuploidy in pig embryos collected 3 days after insemination [24] and only 4.7% aneuploidy in blastocysts collected 5.5 days after insemination [10]. In the last-mentioned study, only 1 out of 62 in vivo blastocysts was triploid and in our study we did not identify triploidy in in vivo blastocysts at all. This may be, in part, because we applied algorithms that detected a greater proportion of chromosome abnormalities (see introduction) and/or it may reflect biological differences between the cohorts studied. Another study also showed lower aneuploidy results with 1.8% triploidy, 1.8% trisomy on chromosome 1 and 10, and 7.9% mosaicism in in vivo porcine embryos at the 2–8 cell stage [29]. These studies used WGA-CGH and one included FISH for aneuploidy screening. In the present study, chromosomal abnormalities were for the first time assessed in porcine embryos using SNP-based algorithms, which can detect aneuploidy more accurately as it allows to identify minor abnormalities such as UPD and polyploidy. Our results therefore suggest that transfer of chromosomally normal blastocysts is recommended for a successful and efficient embryo transfer, whether IVD or IVP.

The study also raises the question of whether the high incidence of aneuploidy in in vivo pig (or indeed any other mammalian) embryos is the natural state of affairs. Early embryo loss has been described to be 30–40% in pigs and cattle [30] and chromosome abnormality is the leading cause of first trimester pregnancy loss in humans [31]. Moreover, are there individuals more prone to generating embryos with chromosome abnormality? In this study, during the in vivo embryo collections, the number of potential embryos derived from each sow ranged from 22 to 50, as based on the number of corpora lutea present on the ovaries, although the average number of total piglets born for this breed is 14.3 piglets. These findings suggest that different levels of losses occur in different sows, and that there is a continuation of losses of embryos and fetuses during development in utero.

### 4.2. IVP Blastocysts Show a High Prevalence of Parthenogenesis

A high incidence of parthenogenetic blastocysts (32.8%) was indicated within our IVP system, which was presented with recombination throughout the chromosome (observed by Karyomapping), except one case that was presented without any recombination. Parthenogenesis was not induced in this study by electrical or chemical stimuli and further research into the cause is therefore of interest as creating parthenotes is undesirable for animal breeding. The parthenogenetic embryos arose from just 3 out of 10 sows in two of the three IVP rounds while cryopreserved sperm from the same boar and ejaculate was used during fertilization, which suggests a possible donor effect. Spontaneous parthenogenetic activation has also been observed in bovine IVP, with 4/61 embryos being parthenogenetic [32]. It has been shown that both induced parthenogenetic and androgenetic embryos develop until day 25 and 30 after embryo transfer in bovine and porcine [33,34]. Normal organ formation was even observed after this initial embryo development, however, the embryos were not viable due to imprinted genes, and they subsequently degenerated. Interestingly, it has been shown that parthenotes can be used to establish a successful pregnancy to deliver a normal piglet at full-term when transferred together with a normal single embryo [35].

### 4.3. A High Incidence of Triploidy in IVP Embryos

Furthermore, we observed a high incidence of triploid embryos (25.0%) in the IVP blastocysts. In porcine IVP, polyspermy has been stated to be a challenge and is shown to be affected by individual boar differences, breed differences, and sperm–oocyte ratio [36,37,38]. Within our IVP system, we previously have reported a polyspermy incidence of 24.8% which was assessed 10–12 h after fertilization, without any difference between six boars [39]. Results in the present study indicate that 12 out of 64 (18.8%) IVP blastocysts were triploid with paternal origin and suggest that fertilization conditions are suboptimal. An aneuploidy incidence of 37% has earlier been reported for in vitro-produced 2-cell embryos [12] and 39.1% in day 6 in vitro blastocysts [25]. Further, maturation could be suboptimal in our IVP system as an increased frequency of chromosomal errors has been reported when porcine oocytes were matured over 40 h compared to an IVM duration of 30–36 h [40]. Additionally, embryo developmental speed and chromosomal abnormality are suggested to be related as the blastocyst percentage from moderate stage embryos (3–4 cells and 5–8 cells) was significantly higher than the rates of delayed (2-cells) and fast stage (>8-cells) embryos on day 2 of culture that showed a higher incidence of chromosomal abnormalities [41]. Only 20.3% of the blastocysts were euploid in the present study, which clearly indicates limitations in our current IVP system. If we would like to use IVP blastocysts for embryo transfer, the system should be optimized to increase the number of euploid blastocysts. Additionally, it would be of interest to study embryos produced with oocytes collected by ovum pick-up rather than oocytes collected from slaughterhouse ovaries, as fewer chromosomal errors are expected.

### 4.4. Origin of Chromosomal Errors

We observed that errors with a paternal origin were more prevalent in IVD embryos, while maternal errors predominated in IVP embryos. This might be explained by the fact that 32.8% of the IVP blastocysts were parthenogenetic. Furthermore, a higher incidence of tri- and hypotriploidy were originating from the sire in both IVD and IVP embryos, which indicate polyspermic fertilization [42,43] as discussed above. All monosomies identified were of maternal origin (i.e., the remaining chromosome was paternal) and this was expected as they typically result from nondisjunction during meiosis in the oocyte. Furthermore, the origin of trisomies was assessed. Trisomies in IVD embryos originated mostly from errors during paternal meiosis I. This is contradicting studies in human and cattle that indicated that most trisomies originate from errors during maternal meiosis I [9,12].

### 4.5. Acrocentric Chromosomes Are More Prone to Non-Disjunction

Our results showed a high incidence of errors on chromosomes 13, 14, and 15 in the IVP blastocysts. In pigs, chromosome 13–18 are acrocentric, which means that the centromere is located near the end of the chromosome. Acrocentric chromosomes seem to be more at risk for non-disjunction events due to the crossing over during the prophase I of meiosis and later during prophase II, as they can suffer premature separation of sister chromatids or reverse segregation [44,45]. Previously, it has also been indicated that the larger porcine acrocentric chromosome 14 and 15 were involved in aneuploidy in in vivo blastocysts [10]. Human acrocentric chromosomes have also been shown to exhibit the highest rate of aneuploidy [44]. Moreover, spontaneous abortions in human are most frequently caused by trisomies [46], especially by trisomy on chromosome 16. It would be of interest to further study arrested pig embryos to see which chromosomal abnormalities on specific chromosomes result in early embryonic arrest.

## 5. Conclusions

Overall, PGT-A discovered a high incidence of aneuploidy and triploidy in IVD and IVP embryos, suggesting that the future application of this technology might improve embryo transfer success in the pig.

## Figures and Tables

**Figure 1 cells-12-00790-f001:**
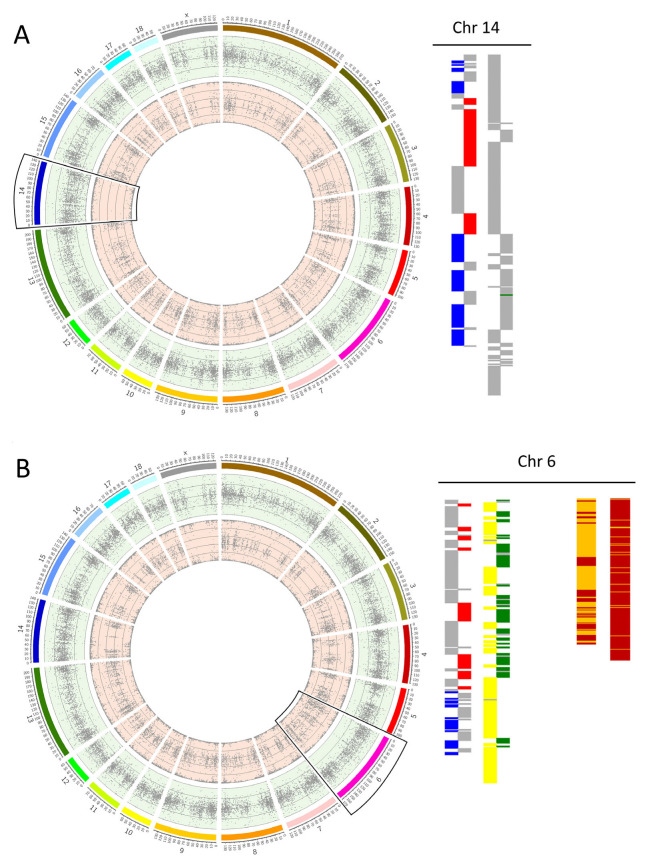
Examples of aneuploidy cases in porcine embryos. (**A**) In vivo embryo with a maternal monosomy on chromosome 14. In the circle graph with BAF and LRR plots, the monosomy is indicated by a lower LRR (green bond) and lack of heterozygosity, shown by missing dots at 0.5 on the BAF (light red bond). The figure on the right shows how the maternal monosomy is demonstrated on the Karyomapping by the presence of haploblocks in the paternal block (blue/red) and the absence of these in the maternal block (where it should be yellow/green, there is only grey blocks—missing of information); (**B**) In vivo embryo identified as having a maternal trisomy on chromosome 6 shown by an increased LRR value and with two inner bonds in the BAF plot (AAB, ABB). On the Karyomapping, the maternal block shows a higher change of haploblocks through the chromosome, which demonstrates the extra information in this chromosome. For trisomies an additional algorithm was applied, the Gabriel–Griffin plot, which in this case showed that this trisomy arose during the Meiosis I.

**Figure 2 cells-12-00790-f002:**
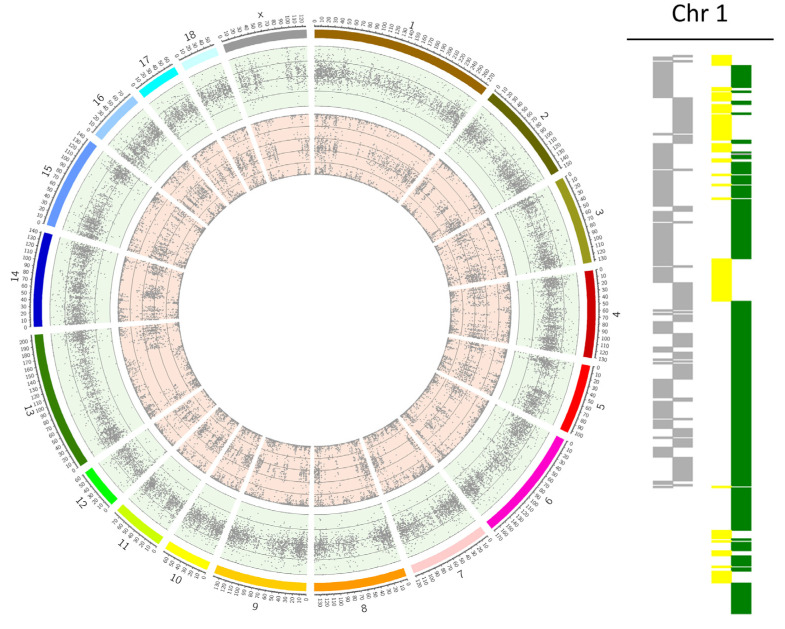
Example of a parthenogenetic case in an in vitro porcine embryo. In the circle graph with BAF and LRR plots, all chromosomes present as disomic with BAF showing plots for the three allele combinations (AA, AB, and BB) and LRR plots around zero. The figure on the right shows how the chromosomes look like all over the genome on the Karyomapping (chromosome 1 was used for example). Here, it is noticeable that there is an absence of haploblocks in the paternal block (where it should be blue/red, there is only grey block—missing information) and the maternal block shows a higher change of haploblocks through the chromosome, which demonstrates the extra information in this chromosome.

**Figure 3 cells-12-00790-f003:**
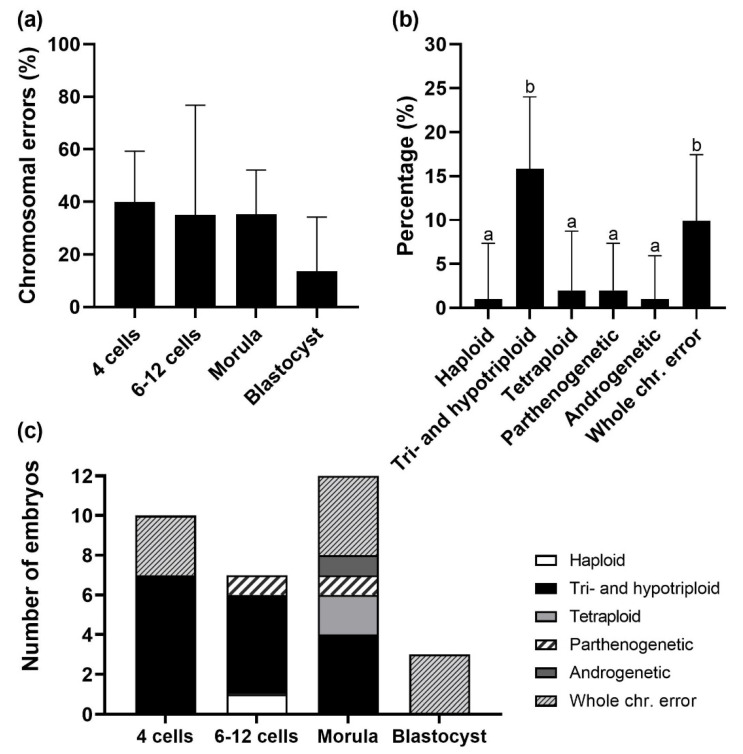
Chromosomal errors in IVD embryos (*n* = 101) as identified by PGT-A. (**a**) Incidence of chromosomal errors at the different embryo developmental stages indicates a trend of fewer errors in blastocysts compared to the 4-cell stage (*p* = 0.056); (**b**) A significantly higher incidence of triploid embryos and embryos with an individual whole chromosome (chr.) error (aneuploidy) was found compared to the other aneuploidy classes; (**c**) Number of embryos for each aneuploidy class per embryo development stage. Data presented as percentage (%) with 95% CI. Bars with different superscripts (a, b) denote significant difference (*p* ≤ 0.05).

**Figure 4 cells-12-00790-f004:**
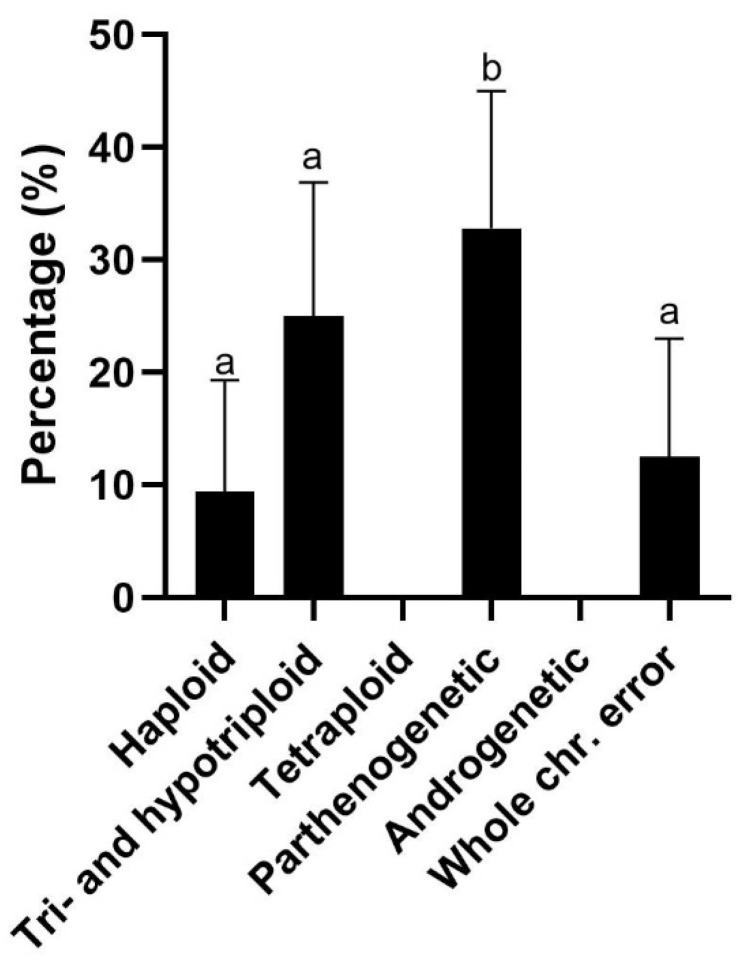
Chromosomal errors in in vitro blastocysts (*n* = 64) per chromosome abnormality class as identified by PGT-A. Data presented as percentage with 95% CI. Bars with different superscripts denote significant difference (*p* ≤ 0.05).

**Figure 5 cells-12-00790-f005:**
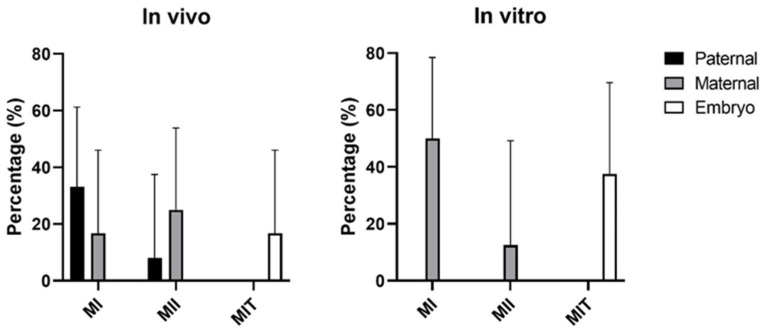
Origin of trisomies (MI; meiosis I, MII; meiosis II, and MIT; mitotic errors during embryonic development) as determined by Gabriel–Griffin plots for IVD and IVP embryos. Data presented as percentage with 95% CI.

**Figure 6 cells-12-00790-f006:**
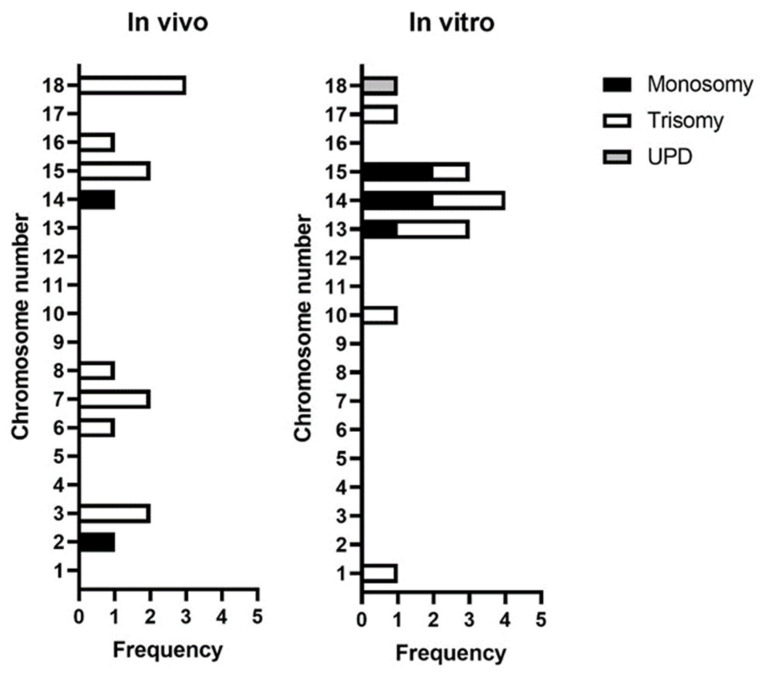
Frequency of whole chromosome errors for in vivo-derived embryos (*n* = 14 whole chromosomal total errors) and in vitro-produced blastocysts (*n* = 14 whole chromosomal total errors).

**Figure 7 cells-12-00790-f007:**
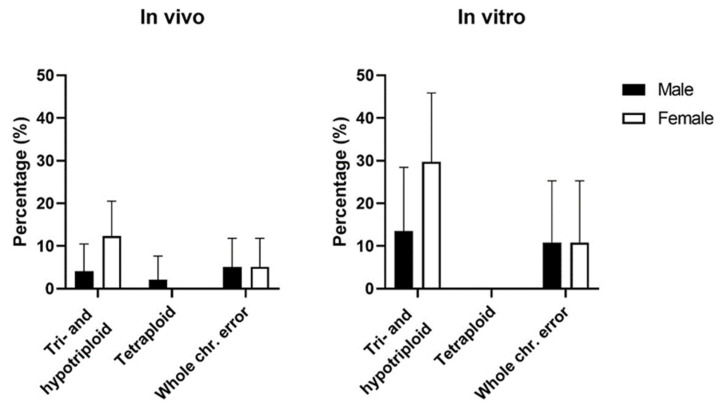
Incidence of chromosome errors based on sex for in vivo-derived embryos (*n* = 97) and in vitro-produced blastocysts (*n* = 37) excluding haploid, parthenogenetic, and androgenetic embryos.

**Table 1 cells-12-00790-t001:** Number of in vivo embryos and their developmental stages collected from five sows on different days of the oestrus cycle. Collection rate (CR) was calculated by dividing the total number of embryos collected by the number of corpora lutea (CLs) counted on both ovaries. Embryos from sow 5 were collected from one horn only and the number of CLs was not recorded.

Sow	Animal	Collection	Embryos	Embryo Stages	CLs	CR
1	Primiparous	Day 4	24	4–8 cells (17 × 4 cell)	26	92%
2	Primiparous	Day 4	15	3–6 cells (equal distr.)	22	68%
3	Nulliparous	Day 5	38	Compacted morulae	50	76%
4	Nulliparous	Day 5	20	Compacted morulae	25	80%
5	Primiparous	Day 6	26	Expanded blastocysts	-	-

**Table 2 cells-12-00790-t002:** Number of blastocysts per sow (S1–S10) according to the aneuploidy classes. Embryos were produced during three IVP rounds which are indicated between borders.

Aneuploidy Classes	S1	S2	S3	S4	S5	S6	S7	S8	S9	S10	Tot
Euploid		2	2	1		2	3	3			13
Haploid		2	1			2				1	6
Tri- and hypotriploid	3	3	3		3	1		1		2	16
Parthenogenetic						4			13	4	21
Whole chr. error	1	3	1	1	1			1			8
Total	4	10	7	2	4	9	3	5	13	7	64

**Table 3 cells-12-00790-t003:** Frequency of chromosomal errors by aneuploidy class and their origin (paternal, maternal, or embryo) for in vivo-derived (IVD) and in vitro-produced (IVP) embryos as identified by PGT-A.

Aneuploidy Classes	Overall		Paternal		Maternal		Embryo	
	IVD	IVP	IVD	IVP	IVD	IVP	IVD	IVP
Haploid	1	6	1			6		
Tri- and hypotriploid	16	16	12	12	4	4		
Tetraploidy	2		*		*			
Parthenogenetic	2	21			2	21		
Androgenetic	1		1					
Whole chr. error	14	14	5	1	7	10	2	2
*Monosomy*	*2*	*5*			*2*	*5*		
*Trisomy*	*12*	*8*	*5*		*5*	*5*	*2*	*3*
*Uniparental disomy*		*1*		*1*				
Total	36	57	19	13	14	41	2	3

* Both parents were involved in the origin of tetraploidy.

**Table 4 cells-12-00790-t004:** Sex ratio and incidence of chromosomal errors for in vivo and in vitro embryos.

	Sex	*n* (%)	% Chromosomal Errors
In vivo	Male	41 (42.3%)	26.8% (11/41)
	Female	56 (57.7%)	30.4% (17/56)
In vitro	Male	18 (48.6%)	50.0% (9/18)
	Female	19 (51.4%)	78.9% (15/19)

## Data Availability

The data that support the findings of this study are available from the corresponding author upon reasonable request.

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
