# Peer review of "Preimplantation Genetic Testing for Aneuploidy (PGT-A) Reveals High Levels of Chromosomal Errors in In Vivo-Derived Pig Embryos, with an Increased Incidence When Produced In Vitro"

_cells, 2023, doi:10.3390/cells12050790_

Round 1

Reviewer 1 Report

The manuscript is interesting in light of the use of the SNP method to analyze aneuploidy in pig embryos. On the other hand, several pieces of information in the manuscript should be clarified before publication.
Specific comments:
Was development till blastocyst stage homogeneous and the same for all embryos and was it 6 days? Usually it doesn`t happen.
The graph showing the number of blastocysts is badly scaled, it is not clear what the exact number is (Fig. 2C). If it is 2 or3 blastocysts, it is to small N for such conclusion.
Discussion:
L305-6: the sentence regarding future research directions is a speculation and should be removed
L324: please clarify the term "normal blastocysts"
Conclusion: please refine it becasuse it is not based on Your results in the current version.

Author Response

Reviewer 1 comments

Response

Was development till blastocyst stage homogeneous and the same for all embryos and was it 6 days? Usually, it doesn`t happen.

Thank you for your comments and suggestions. This is indeed true, a few blastocysts from one sow were already collected on day 5. For the other sows we only observed and collected blastocysts on day 6. We clarified this now by adjusting L139 to ‘day 5 or 6 when they reached blastocyst stage.’

The graph showing the number of blastocysts is badly scaled, it is not clear what the exact number is (Fig. 2C). If it is 2 or 3 blastocysts, it is to small N for such conclusion.

We changed the scaling and size of this figure so that this is possible to assess.

L305-6: the sentence regarding future research directions is a speculation and should be removed

We adjusted this sentence as we believe that further studies in porcine and bovine could give meaningful insights.

L324: please clarify the term "normal blastocysts"

Here we used the term ‘Chromosomally normal blastocysts' to clarify that transfer of embryos with chromosomal abnormalities should be avoided.

Conclusion: please refine it because it is not based on your results in the current version.

We agree, part of the conclusion is moved to the discussion to come to a clear conclusion.

Reviewer 2 Report

In manuscript cells-2202061 the authors report an analysis of the karyotypes of pig embryos either during in vitro production (IVP) or in vivo derived (IVD) using single nucleotide polymorphism (SNP) based PGT-A (preimplantation genetic testing for aneuploidy) algorithms. The work is mainly aimed at identifying the causes of genome alterations and possible ways to improve porcine embryo transfer success.

Overall, the work has some specific point of interest, and the IVD vs. IVP effects on chromosomal asset in embryos is interesting. However, the manuscript has a major flaw in the reported numbers, which make the results no more than preliminary. Error bars are very prominent and only a minor amount of data passes the significant threshold of 0.05 of the p-value. As such, this reviewer recognizes the good work performed so far, and invites the authors to collect additional data to make the manuscript statistically robust and suitable for publication for a future submission.

Specific points to be checked:

Section 2.7. The meaning of the statement is unclear. Are these data available, or not? If yes, where, and how?

Section 3.1. The trend reported for data in figure 2a is not evident. The first three bars are statistically identical, and the fourth bar alone – which also lack statistical significance, even if by a few decimal points – cannot justify the identification of a ‘trend’. In addition, as also reported in the Discussion, this result is expected, as older embryos are more prone to die in case of karyotype anomalies. Thus, this is not an original finding.

Section 3.3. Only one datum passes the statistical threshold; all the others are not significant. Thus, most section 3.3 is just speculative and not supported by data. For this reason, Table 4 of the manuscript is misleading for the reader because it reports data not supported by the statistics. Consequently, this also impairs the meaning of section 4.5 in the Discussion.

Section 4.1. The last few lines of the section are not clear to me – the authors should better explain what they mean, especially in the last 2-3 lines; what is the message to the reader?

Section 4.3. Also here, the final part of the section is not completely clear, the authors should better explain what the message is they would like to leave to the reader.

Section 4.4. A deeper discussion (a few lines) about acrocentric chromosomes behavior in mammalian meiosis would be appreciated by the reader.

Author Response

Reviewer 2 comments

Section 2.7. The meaning of the statement is unclear. Are these data available, or not? If yes, where, and how?

Thank you for your comments and suggestions.

We clarified now that this data is not publicly available.

The trend reported for data in figure 2a is not evident. The first three bars are statistically identical, and the fourth bar alone – which also lack statistical significance, even if by a few decimal points – cannot justify the identification of a ‘trend’. In addition, as also reported in the Discussion, this result is expected, as older embryos are more prone to die in case of karyotype anomalies. Thus, this is not an original finding.

This was a pilot study to assess if using SNP data for PGT-A works for pig embryos. We are working towards expanding the study to increase the sample size. We believe it is important to show the results we obtained. It is not an original finding, but in line what others have reported before.

Section 3.3. Only one datum passes the statistical threshold; all the others are not significant. Thus, most section 3.3 is just speculative and not supported by data. For this reason, Table 4 of the manuscript is misleading for the reader because it reports data not supported by the statistics. Consequently, this also impairs the meaning of section 4.5 in the Discussion.

We agree with this and changed the results in L270-273. We also realize that section 4.5 in the discussion section is not relevant anymore with the current results and removed this from the discussion.

Section 4.1. The last few lines of the section are not clear to me – the authors should better explain what they mean, especially in the last 2-3 lines; what is the message to the reader?

We agree and we removed the last lines as our conclusions are already made in the lines before.

Section 4.3. Also here, the final part of the section is not completely clear, the authors should better explain what the message is they would like to leave to the reader

We clarified the last part of section 4.3

Section 4.4. A deeper discussion (a few lines) about acrocentric chromosomes behavior in mammalian meiosis would be appreciated by the reader.

More information about acrocentric chromosomes is now included in section 4.4

Reviewer 3 Report

The presented manuscript concerns the results of the chromosomal error tests (PGT-A) in the preimplantation in vivo- or in vitro -derived porcine embryos. The results are interesting and indicated the high incidence of chromosomal abnormalities, which may be a cause of the low success in the in vitro procedure in pigs. I have only a few doubts or suggestions.

Introduction -lines 62-67 –“The first allows to visualize chromosomes…..” theses information is additional, not strictly connected with the manuscript topic, and can be removed.

In sentence (Line 75) the reference regarding “as in humans” is missing.

Quality of fig.1, the improvement of the quality of this figure is needed. It is very difficult to read. I wonder if the Authors could prepare a similar figure for the in vitro embryos.

M&M – the references to how the embryos were qualified.  Line 90 – why Authors wrote that females were at day 4,5 and 6 days of the estrous cycle. If the embryos were obtained, in my opinion, it should be an early pregnancy, not the estrous cycle.

The sources of the PZM-5 and PXM and other reagents should be indicated. Shortcuts should be explained when they appeared for the first time in the text.

Line 335-338- literature ?? – references missing.

Author Response

Reviewer 3 comments

Response

Introduction -lines 62-67 –“The first allows to visualize chromosomes…..” theses information is additional, not strictly connected with the manuscript topic, and can be removed.

Thank you for your comments and suggestions. Line 62-67 explains the benefits of using SNP based PGT-A algorithms compared to techniques used in other studies. We find that it is important to include this information in the introduction as this is a pilot study to test out if using SNP based algorithms work and we come back to this in the discussion.

In sentence (Line 75) the reference regarding “as in humans” is missing.

The reference is now included.

Quality of fig.1, the improvement of the quality of this figure is needed. It is very difficult to read. I wonder if the Authors could prepare a similar figure for the in vitro embryos.

We changed the figure to improve the quality and size of the plots. Figures for in vivo and in vitro embryos look the same so it is not needed to show both. We also added an example of a parthenogenetic IVP embryo as one of the other reviewers suggested including this.

M&M – the references to how the embryos were qualified.  Line 90 – why Authors wrote that females were at day 4,5 and 6 days of the estrous cycle. If the embryos were obtained, in my opinion, it should be an early pregnancy, not the estrous cycle.

Day 0 is the onset of estrous, so day 4, 5 and 6 are indeed early pregnancy. This is also defined in other articles that collected in vivo embryos. We removed estrous and hope this is clearer now.

The sources of the PZM-5 and PXM and other reagents should be indicated. Shortcuts should be explained when they appeared for the first time in the text.

Yes, thank you. This is adjusted now in L103-106.

Line 335-338- literature ?? – references missing.

The reference is now included.

Reviewer 4 Report

The manuscript “Preimplantation genetic testing for aneuploidy (PGT-A) reveals high levels of chromosomal errors in in vivo derived pig embryos, with an increased incidence when in vitro produced” by Reina Jochems, Carla Canedo-Ribeiro, Giuseppe Silvestri, Martijn F.L. Derks, Hanne Hamland, Louisa J. Zak, Egbert Knol, Alan H. Handyside, Eli Grindflek and Darren K. Griffin of interest to the professional community as it opens up methods that have been carefully developed for working with human embryos to applications for farm animals and other model organisms. Although this work is not pioneering, it provides some new evidence about the level of chromosomal instability in early mammalian embryogenesis and therefore deserves to be published in the journal Cells after a minor revision.

Important note:

Lines 200-203. The results on the identification of parthenogenetic embryos are of great interest, so for these embryos it is necessary to present the results of karyomapping and prove that this is not the result of haploid cell division. In any case, the result described is very important, but it is necessary to discuss in more depth the nature of the origin of such embryos (show the level of recombination for gynogenetic embryos, if it exists or show its absence).

Lines 342-354. When discussing the potential for the development of parthenogenetic embryos, information on the origin of the diploid set of the discovered androgenetic embryo would also be important: whether there was fertilization by two spermatozoa or whether this was a consequence of the duplication of a single paternal genome.

Additional comment:

Lines 356-359. The presence of a large proportion of triploid IVP embryos may indicate suboptimal fertilization conditions causing polyspermy, which may be related to the transfer of technologies optimized for human embryos, this needs to be discussed.

Author Response

Reviewer 4 comments

Response

Lines 200-203. The results on the identification of parthenogenetic embryos are of great interest, so for these embryos it is necessary to present the results of karyomapping and prove that this is not the result of haploid cell division. In any case, the result described is very important, but it is necessary to discuss in more depth the nature of the origin of such embryos (show the level of recombination for gynogenetic embryos, if it exists or show its absence).

Thank you for your comments and suggestions. An image of a parthenogenetic embryo has been added with the graph and the karyomapping example for one chromosome in Figure 2. Additionally, more info was added to the discussion 4.2 section.

Lines 342-354. When discussing the potential for the development of parthenogenetic embryos, information on the origin of the diploid set of the discovered androgenetic embryo would also be important: whether there was fertilization by two spermatozoa or whether this was a consequence of the duplication of a single paternal genome.

We also find this very interesting. Our data suggest that it is a dispermic event, so two sperm cells fertilized an empty egg. This has been added in the results section under 3.2. As we only observed one androgenetic embryo, we decided not to further discuss this in the discussion.

Lines 356-359. The presence of a large proportion of triploid IVP embryos may indicate suboptimal fertilization conditions causing polyspermy, which may be related to the transfer of technologies optimized for human embryos, this needs to be discussed.

Yes, this was mentioned later in lined 376-382, but we now moved this information together under the discussion session 4.5 and additionally clarified it.

Round 2

Reviewer 2 Report

This Reviewer appreciate the efforts of the Authors to improve the overall quality of the manuscript. No major flaws are evident now.